# Bicontinuous Cubic Liquid Crystals as Potential Matrices for Non-Invasive Topical Sampling of Low-Molecular-Weight Biomarkers

**DOI:** 10.3390/pharmaceutics15082031

**Published:** 2023-07-28

**Authors:** Maxim Morin, Sebastian Björklund, Emelie J. Nilsson, Johan Engblom

**Affiliations:** 1Biofilms—Research Center for Biointerfaces, Malmö University, SE-205 06 Malmö, Swedensebastian.bjorklund@mau.se (S.B.); emelie.nilsson@mau.se (E.J.N.); 2Department of Biomedical Science, Faculty of Health and Society, Malmö University, SE-205 06 Malmö, Sweden

**Keywords:** tryptophan, kynurenine, tryptophan-to-kynurenine ratio, cancer-related biomarkers, non-invasive extraction, bicontinuous cubic liquid crystal, bilayer partitioning, glycerol monooleate, DOTAP, X-ray diffraction, humidity scanning (HS) QCM-D

## Abstract

Many skin disorders, including cancer, have inflammatory components. The non-invasive detection of related biomarkers could therefore be highly valuable for both diagnosis and follow up on the effect of treatment. This study targets the extraction of tryptophan (Trp) and its metabolite kynurenine (Kyn), two compounds associated with several inflammatory skin disorders. We furthermore hypothesize that lipid-based bicontinuous cubic liquid crystals could be efficient extraction matrices. They comprise a large interfacial area separating interconnected polar and apolar domains, allowing them to accommodate solutes with various properties. We concluded, using the extensively studied GMO-water system as test-platform, that the hydrophilic Kyn and Trp favored the cubic phase over water and revealed a preference for locating at the lipid–water interface. The interfacial area per unit volume of the matrix, as well as the incorporation of ionic molecules at the lipid–water interface, can be used to optimize the extraction of solutes with specific physicochemical characteristics. We also observed that the cubic phases formed at rather extreme water activities (>0.9) and that wearing them resulted in efficient hydration and increased permeability of the skin. Evidently, bicontinuous cubic liquid crystals constitute a promising and versatile platform for non-invasive extraction of biomarkers through skin, as well as for transdermal drug delivery.

## 1. Introduction

The detection of cancer at an early stage is highly important as it has a significant impact on patient survival. Currently, skin cancer diagnosis relies on visual evaluation of the suspected lesion, often followed by a tissue biopsy. The accuracy of the visual diagnosis varies significantly (49–81%) depending on, e.g., the clinician’s experience and the characteristics of the lesion [1,2,3,4,5]. Benign nevus is one example of a very common but harmless lesion which may be mistaken for cutaneous melanoma, thus resulting in numerous unnecessarily excised benign lesions. Therefore, the development of alternative or complementary non-invasive methods for early stage skin cancer diagnosis based on, e.g., the extraction of endogenous biomarkers, is highly desirable. It has furthermore been shown that sustained inflammation sometimes acts as a precursor for cancer. For example, actinic keratoses and Bowen’s disease are precursors for squamous-cell carcinoma [6]. Thus, the detection of inflammation biomarkers can also serve as an early warning for disease onset [7]. As a result, from recent progress in cancer research, there is an emerging number of endogenous substances associated with inflammation and cancer to choose between, e.g., NF-κB, IL-6, IFN-γ, TNF-α, enzyme indoleamine-2,3-dioxygenase (IDO), and BRAF gene mutations [8,9,10]. Most of these substances are, however, too big (i.e., >500 Da) to easily penetrate from the viable epidermis through the *stratum corneum* to appear on the skin surface [11,12]. Instead, we targeted low-molecular-weight (LMW) compounds present in viable skin, where, e.g., the catabolism of the essential amino acid tryptophan (Trp) along the kynurenine pathway (KP) has been reported to play a crucial role in regulation of the immune response [13,14]. The conversion of Trp to kynurenine (Kyn) is catalyzed by the enzymes indolamine 2,3-dioxygenase 1 and 2 (IDO-1/IDO-2) and tryptophan 2,3-dioxygenase (TDO) [15]. At abnormal conditions, the expression of IDO-1 is upregulated, leading to the increased conversion of Trp to Kyn. Moreover, it has previously been confirmed that the change in the Trp/Kyn ratio is associated with several diseases, including cancer [16,17,18,19].

The intrinsic physicochemical properties of Trp (204.2 Da, log D_o/w_ –1.1 at pH 7.4, Chemicalize v. 19.7.0, 2019, ChemAxon) and Kyn (208.2 Da, log D_o/w_ –1.9 at pH 7.4, Chemicalize v. 19.7.0, 2019, ChemAxon) make them ideal candidates for non-invasive topical sampling. However, their endogenous concentrations are low, and successful non-invasive detection rely on high permeability in the skin and a high partitioning from the skin to the intended extraction matrix. Whereas hydrating the tissue favors skin permeability for a wide range of compounds [20,21], it is more challenging to find a versatile matrix. Lipid-based liquid crystals, with their rich aqueous phase behavior, offer an interesting opportunity, in particular, bicontinuous cubic liquid crystals [22,23]. These lyotropic cubic phases comprise a large interfacial area separating their interconnected polar aqueous and apolar lipid domains, which makes them susceptible to accommodate various types of solutes, being hydrophilic as well as amphiphilic or lipophilic [24,25]. Bicontinuous cubic phases may be formed by polar lipids, such as some mono- and diglycerides, or phospholipids, alone or in combination, in contact with water. Some examples comprise glyceryl monooleate, glyceryl monoelaidate, glyceryl monolinoleate, glyceryl dioleate, dioleoyl phosphatidylglycerol, distearoyl phosphatidylglycerol, dioleoyl phosphatidylethanolamine, dioleoyl phosphatidylserine, and phytantriol [26,27,28]. By far, the most studied polar lipid forming cubic phases is glyceryl monooleate (GMO), whereas the more chemically inert phytantriol has emerged as an attractive alternative for some applications [29,30]. The GMO-water system forms two subsequent cubic phases on hydration, i.e., first the gyroid (Ia3d) and then the double diamond cubic phase (Pn3m), while adding anionic or zwitterionic lipids to this system has been shown to induce the formation of the third and highly swollen primitive cubic phase (Im3m) [31,32].

Furthermore, cubic phases have as such been extensively explored as potential drug delivery systems for a wide range of drugs with varying solubilities in water, ranging from small molecules (such as acetyl salicylic acid, tocopherol, metronidazole, tetracycline, timolol maleate, cephazolin, pindolol, propranolol, pyrimethamine) to peptides and proteins (e.g., desmopressin, lysine vasopressin, somatostatin, insulin, lysozyme, hemoglobin, ovalbumin, bovine serum albumin) [27,33,34,35]. Clogston and Caffrey [36] reported on the drug release of Trp (approximately 10 mM initial concentration, corresponding to 0.2% (*w*/*w*)) from the double diamond cubic phase (Pn3m), while Glatter and coworkers [37] investigated the effect on lipid phase behavior in a very similar setting with larger amounts of Trp (up to 4% (*w*/*w*)). They found that larger amounts of Trp affected the lattice parameter of the cubic phase and that amino acids with hydrophobic parts may locate within the lipid–water interface. These are very encouraging results as amino acids apparently show affinity for lipid bilayers. However, to the best of our knowledge, to date, no one has attempted to use lipid based liquid crystals, and bicontinuous cubic phases in particular, for the extraction of endogenous molecules from the skin’s surface.

The aim of the current project was therefore to explore the versatility of bicontinuous cubic liquid crystals as potential matrices for the non-invasive topical sampling of LMW biomarkers, i.e., Trp and Kyn. We adopted the extensively studied nonionic glycerol monooleate (GMO)-water system as the base and introduced a structurally related cationic lipid, dioleoyl trimethylammonium propane (DOTAP), to moderate the cubic structure. Of particular interest is how the incorporation of charged lipids and addition of electrolytes can affect the phase behavior and, in turn, the interfacial area in the cubic unit cell and subsequent biomarker partitioning to the lipid bilayer.

## 2. Materials and Methods

### 2.1. Materials

Glyceryl monooleate (GMO, RYLO^TM^ MG 19 Pharma) was kindly provided by Danisco Cultor (Brabrand, Denmark), and 1,2-dioleoyl-3-trimethyl-ammonium-propane (DOTAP) was purchased from Avanti Polar Lipids Inc. (Alabaster, AL, USA). Both lipids were used without further purification. L-tryptophan (Trp), L-kynurenine (Kyn), NaCl, and LiCl were obtained from Sigma-Aldrich (St. Louis, MO, USA), NaH_2_PO_4_·H_2_O was sourced from Merck (Darmstadt, Germany), and ethanol and methanol (both > 99.8% (*v*/*v*)) were purchased from VWR International (Fontenay-sous-Bois, France). The water used for sample preparation was of Milli-Q grade (18.2 MΩ·cm).

### 2.2. Sample Preparation

*GMO:H_2_O:* Solid GMO (0.1 g) was weighted in 1.7 mL glass vials and melted at 40 °C by immersing the vials into a water bath. After re-crystallization of the GMO, the desired amount of MQ water was added to the vials at room temperature (21 ± 0.3 °C). The vials with GMO:H_2_O were then centrifuged 6 times for 5 min at 1000× *g* and subsequently kept in the dark until fully equilibrated (≥1 week).

*GMO:DOTAP:H_2_O:* Samples were prepared by mixing appropriate amounts of GMO and DOTAP dissolved in ethanol in 1.7 mL glass vials. The solvent was evaporated using a GeneVac system at 35 °C (EZ-2 Plus Evaporating System, Genevac LTD, Ipswich, UK) and samples were then further dried under a vacuum overnight. Two parallel sets of samples were prepared by adding the desired amount of either Milli-Q water or 150 mM NaCl (aq) solution. The vials were then centrifuged 6 times for 5 min at 1000× *g* and subsequently kept in the dark until fully equilibrated.

*LiCl (sat. aq):* Saturated LiCl solution for humidity scanning QCM-D experiments was prepared by mixing excess amounts of LiCl salt in water for several days and filtering the final saturated solution twice to remove excess solid LiCl.

### 2.3. Small-Angle X-ray Diffraction (SAXD)

Small-angle X-ray diffraction (SAXD) was performed on a Xeuss 3.0 SAXS/WAXS laboratory-based instrument (Xenocs, Grenoble, France) at Malmö University (Malmö, Sweden). The X-ray beam was generated by a Cu K_α_ source (λ = 1.541 Å). All samples were kept in an ambient atmosphere during measurements at 25 °C using a temperature-controlled Peltier gel-holder stage. The gel-holder utilized an O-ring as a spacer between two Kapton films (DuPont^TM^ Kapton^®^, 0.013 mm thickness, Goodfellow, UK), sealed in between two metal plates with a 5 mm opening. The diffraction data were collected by a Pilatus3 R 300K hybrid photon counting detector at two different sample-to-detector distances (STDD) of 800 mm and 1700 mm. These two STDDs covered the *q*-range 0.0002 ≤ *q* (Å^−1^) ≤ 0.36, where *q* is the scattering vector and is defined as:(1)$$\left|q\right|=q=\frac{4\pi }{\lambda }\sin\left(\frac{\theta }{2}\right)$$
where *θ* is the scattering angle. The *q* scale was calibrated using silver behenate. One-dimensional (1D) data were obtained by the azimuthal averaging of 2D-diffraction pattern recorded by the detector, and the data were corrected for background scattering and normalized to the direct beam using the Xenocs XSACT software (version 2.6). The exposure time was 30 min for each sample at each STDD.

### 2.4. Humidity Scanning (HS) QCM-D

The hydration of the lipid films was investigated using humidity scanning QCM-D [38]. The technique, besides being a highly accurate for determination of a mass of materials adsorbed on piezoelectric quartz sensor based on Sauerbrey methodology [39], also provides information about the viscoelastic properties of the adsorbed material via the dissipation data [40]. It works by applying an oscillating potential on a quartz crystal and monitoring the frequency of the resulting oscillating shear motion, which generates an acoustic wave. The resonance condition occurs when the wavelength of the resulting acoustic wave is an odd integer of the quartz sensor’s thickness. The information of the mass of the adsorbed material is obtained from the resonance frequency. The mass of the adsorbed material can be determined using the Sauerbrey equation (Equation (2)) [39] under the assumptions that the mass of the material is small compared to the mass of the crystal and that the material is rigidly adsorbed and homogenously distributed over the active area of the crystal.
(2)$$-\frac{\Delta f}{n}=\frac{2f_{0}^{2}m_{f}}{Z_{q}}$$

The Sauerbrey equation describes the relationship between the negative frequency change Δ*f*, normalized per overtone *n*, and the product of the areal film *m*_f_ (kg·m^−2^) and the fundamental resonance frequency *f*_0_ of the quartz sensor (~5 MHz) normalized by the acoustic impedance of quartz *Z_q_* (8.8 × 10^6^ kg·m^−2^·s^−1^). In addition to the areal masses of the films obtained from the QCM-D experiments, the films are also described by their estimated thicknesses. The thickness of a dry film, *d*, can be calculated from the areal mass of the dry film *d* = *m*_f_/*ρ*, where *ρ* is the density of the dry film. The density of a dry lipid film, constituting mostly of GMO in this work, was assumed to be 0.94 g·cm^−3^. However, it is important to note that this was only an estimation of the film thickness. As mentioned earlier, the QCM-D technique also monitors the dissipation, *D,* which is related to the decay time of the oscillating resonator when the alternating potential is turned off. The viscoelastic properties of the film adsorbed on the quartz crystal have a strong impact on its dissipation energy, which is related to the decay time. Therefore, dissipation provides information about the rheological properties of the film as well as complementary data during the hydration process.

A q-sense QCM-D E4 with humidity module QHM 401 and AT-cut SiO_2_ (QSX 303, 5 MHz) sensors from Biolin Scientific AB (Gothenburg, Sweden) were used in this work. The humidity module was equipped with a Gore membrane, which separated the flowing solution from the sensor, allowing only the water vapors from the solution to diffuse across the membrane and regulate the RH above the film coated on the surface. New sensors were gently washed with ethanol and Milli-Q water and dried by the flow of nitrogen, while used sensors were cleaned according to the cleaning protocol described in the q-sense guidelines manual (cleaning protocols B for QSX 303). No difference was observed in the measurements performed with new and reused sensors.

Lipids (GMO and DOTAP) were dissolved in ethanol in appropriate ratios so that the final concentration was 8 mM. The humidity scanning QCM-D experiment was initiated by measuring the uncoated sensor in a dry N_2_ atmosphere at 25 °C. After that, sensors were coated with a lipid film by spin-coating, where 10–20 µL of lipid solution was applied once on the surface of the sensor. In a study by Björklund and Kocherbitov [41], it was found that film thickness was primarily dependent on the concentration and not on the number of solution applications. The coated sensors were then dried overnight in a vacuum and then placed back into the humidity module. The measurements were initiated by flowing dry N_2_ gas until a stable baseline was observed (usually 30 min). After that, N_2_ gas flow was stopped, and the hydration experiment was performed according to a procedure described in detail elsewhere [38]. In brief, the measurement relied on the continuous and controlled dilution of a saturated LiCl solution that was flowing through the humidity chamber. Since only water vapor could pass across the Gore membrane, the RH above the sensor was continuously regulated by adjusting the water activity, *a*_w_, of the LiCl solution (*a*_w_ = RH/100%).

### 2.5. Swelling of Bicontinuous Liquid Crystalline Aqueous Phases

The swelling laws previously disclosed by Engblom and Hyde [42] for lyotropic liquid crystals were used together with X-ray data to verify structure determinations and more accurately identify the phase boarders and swelling limits. These equations further allowed for a more in-depth analysis of the internal geometries of individual phases, such as the lipid/water interfacial area per unit cell and how this relates to macroscopic volumes, water channel radii and lengths, etc.

According to Engblom and Hyde, the lattice parameter, *a* (see Appendix A for calculations of *a* from diffraction data [43,44]), can be derived for reverse bicontinuous cubic phases using Equation (3):(3)$$a=l{\left(\frac{-2\pi \chi }{H}\right)}^{1/3}{\left[\sqrt{3}\sin\left(\frac{\Delta }{3}\right)+\cos\left(\frac{\Delta }{3}\right)\right]}^{-1}$$
where l is the average lipid monolayer thickness (here, 17 Å); χ is a topology index of the surface, known as Euler-Poincaré characteristics (–8 for Ia3d, –2 for Pn3m, and –4 for Im3m); and *H* is a dimensionless characteristic, the “homogeneity index,” which combines the surface-to-volume ratio of a hyperbolic surface with its topology (0.7665 for Ia3d, 0.7498 for Pn3m, and 0.7163 for Im3m). Δ is defined as,
(4)$$\Delta =\tan^{-1}\left(\frac{\sqrt{1-\Phi_{l}^{2}}}{-\Phi_{l}}\right)$$
where $\Phi_{l}=1-\Phi_{w}$ is the volume fraction of the lipid, and $\Phi_{w}$ the water volume fraction. The water volume fraction can then be recalculated into weight fraction of water as:(5)$$\varphi_{w}=\frac{1}{1+\rho_{l}\frac{1-\Phi_{w}}{\Phi_{w}}}$$
where $\rho_{l}$ is the density of the lipid (0.94 g·cm^−3^).

### 2.6. Partitioning of Trp and Kyn into Lipid Bilayer

The partition experiment of Trp and Kyn into the lipid bilayer of a cubic phase was performed in accordance with the method described by Engström et al. [45], where the authors investigated the lipid bilayer/water partition of model drug clomethiazole. Four concentrations of Trp and Kyn in the range from 0.125 mM to 1 mM, corresponding to the lipid:Trp(Kyn) ratio in the range 7200:1–900:1, were prepared in Milli-Q water in order to investigate the effect of concertation on the partitioning. The fully swollen cubic phases (~0.25 g) were prepared by mixing the appropriate amount of GMO with an excess of Milli-Q water (1:1 weight ratio) in 1.7 mL glass vials and left to equilibrate for 7 days. When the samples had equilibrated, excess water was removed and replaced with 500 µL of aqueous solution with different concentrations of Trp/Kyn (molar ratio 1:1). The first sampling was performed after one week of equilibration by withdrawing 50 µL of the aqueous phase. The second sampling was performed in an identical manner after 2 weeks of equilibration to investigate the time aspect on the partitioning. All samples collected during the partition study were diluted to a final volume 500 µL with Milli-Q water and filtered with 0.2 µm syringe filters (13 mm PTFE membrane, VWR International, Radnor, PA, USA) prior to the HPLC-UV analysis. All partition experiments were performed in triplicate.

### 2.7. Bilayer Partition Coefficient

The cubic liquid-crystalline phase consists of two domains: a lipid bilayer domain and a water domain. The calculation of the lipid bilayer/water partition coefficient, *K*_bl/w_, was described by Engström et al. [45]. Briefly, their work states that the partition coefficient is defined as:(6)$$K_{\mathrm{bl}/w}=\frac{{\left[X\right]}_{\mathrm{bl}}}{{\left[X\right]}_{w}}$$
where [*X*] is the concentration of an analyte of interest (e.g., Trp and Kyn) in the bilayer (bl) and in water (w). Whereas the concentration of a substance in the aqueous phase can easily be determined by a suitable analytical procedure (e.g., HPLC, LC-MS etc.), the determination of the concertation in the bilayer requires two assumptions. The first assumption is based on the fact that the GMO has very low solubility in water, around 10^−6^ M [46] with an overall HLB of 3.8 [47], implying that there is no free GMO existing in the aqueous phase (i.e., all GMO makes up the lipid bilayer). The second assumption is that the concentration of an analyte in the water channels of the cubic phase is the same as in the water bulk phase, which is based on the equilibrium between chemical potentials of analytes in water channels and in the bulk phase. Thus, the partition coefficient can be calculated by rewriting Equation (7) as follows:(7)$$K_{\mathrm{bl}/w}=(V_{w}\left({\left[X\right]}_{0w}-{\left[X\right]}_{w}\right)/{\left[X\right]}_{w}-V_{w,\mathrm{cube}})/V_{\mathrm{bl}}$$
where *V*_w_ is the volume of a water solution containing analyte *X* added to the cubic phase, [*X*]_0w_ is the initial concentration of an analyte, [*X*]_w_ is the concentration of analyte in the water phase after equilibration, *V*_bl_ is the volume of GMO, and *V*_w_*,*_cube_ is the volume of water that was added to GMO to form a cubic phase. The cubic phase/water partition coefficient, *K*_Q/w_, can then be obtained from the expression disclosed as Equation (8):(8)$$K_{Q/w}=1+(K_{\mathrm{bl}/w}-1)\frac{V_{\mathrm{bl}}}{\left(V_{w,\;\mathrm{cube}}+V_{\mathrm{bl}}\right)}$$

## 2.8. HPLC-UV Analysis

The quantification of Trp and Kyn was performed by the HPLC-UV system (Agilent 1100 Series, Waldbronn, Germany). The chromatographic separation of Trp and Kyn was carried out on a 250 mm × 4.6 mm Kromasil C18 column with particle size of 5 µm (AkzoNobel, Bellefonte, PA, USA). Analytes were separated by gradient elution using mobile-phase A consisting of 10 mM NaH_2_PO_4_ (pH 2.8) and mobile-phase B consisting of 100% MeOH at 0.9 mL/min flow rate and 40 °C column temperature. The gradient profile was adopted from previous work [48] and modified as follows: mobile-phase B was kept at 25% for 7 min; then, phase B was gradually increased to 95% over 4 min and kept at 95% for 4 min, after which phase B was decreased to 25% over 0.1 min and kept at 25% for the final 1.9 min, resulting in a total run time of 17 min. The injection volume was set to 20 µL. The detection of Trp and Kyn was performed at their UV absorbance maxima, at 280 nm and 360 nm, respectively. Stock solutions of 10 mM of Trp and Kyn for calibration curve were prepared in Milli-Q water and kept in the freezer (−20 °C) for no longer than one day after preparation. Calibration standards for the calibration curve were analyzed in the range from 0.78 µM to 100 µM (R^2^ > 0.999) the same day as the experimental samples. The quantity of analytes was determined by manual integration of the corresponding peaks using OpenLAB software (Lab Advisor Basic Software, Agilent, Germany). The concentrations of Trp and Kyn in the unknown samples were determined using the calibration curve obtained from standards solutions. The LOQ for Trp and Kyn were determined to 0.43 µM and 0.69 µM, respectively (see Appendix A for further information regarding the LOD, LOQ, accuracy, and precision of the HPLC-UV method, Appendix A).

## 3. Results

### 3.1. The Matrix

Due to their specific physicochemical properties, bicontinuous cubic lipid-based liquid crystals have a high potential to be used as matrices for the non-invasive topical sampling of low-molecular-weight biomarkers. In this work, we adopted the extensively studied GMO-water system as our model and introduced a structurally related cationic lipid (DOTAP) with the dual purpose to increase water swelling and to obtain a charged lipid–water interface. GMO alone forms a reversed micellar (L_2_) phase at 25 °C and low water content, which, on further hydration, first transitions into a lamellar liquid crystal (L_α_), and then to two subsequent reversed types of bicontinuous cubic phases—first a Gyroid cubic phase (C_G_, space group Ia3d) and then a Double Diamond cubic phase (C_D_, space group Pn3m), which has a limited swelling of about 40% (*w*/*w*) and co-exists with excess water [49,50,51]. DOTAP, on the other hand, only forms a lamellar phase (L_α_), which swells up to about 95% (*w*/*w*) water, corresponding to a lattice parameter *a,* of 708 Å (see Appendix A). The further addition of water most probably results in the formation of fully hydrated unilamellar vesicles dispersed in excess water, which has previously been observed for other charged lipids [52]. Figure 1 shows a part of the phase diagram for the ternary system GMO/DOTAP/H_2_O, outlined based on visual inspection of the samples between crossed polarized windows, polarized light optical microscopy (PLOM), and small-angle X-ray diffraction (SAXD) (see Appendix A). Phase boarders and swelling limits were further determined by comparing the lattice parameters obtained by SAXD with the expected dimensions estimated from the appropriate swelling laws previously disclosed by Engblom and Hyde. These laws are also briefly outlined in the Materials and Methods section above, and the results are depicted in Figure 2 (also see Appendix A) [31,42]. SAXD data and corresponding calculated dimensions are summarized in Table 1 (see also Appendix A).

At low water content (up to about 30% (*w*/*w*)), there was no significant effect on the phase behavior from adding the positively charged DOTAP (within the range from GMO/DOTAP 100/0 to 80/20 (*w*/*w*)) (see Appendix A). The transition from L_α_ to cubic C_G_ occurred at approximately the same water content, independently of the amount of DOTAP added, while the C_G_ swelled extensively from 30% (*w*/*w*) water (*a* = 126.8 Å) in the pure system to 45% (*w*/*w*), with the highest amount of DOTAP (*a* = 184.3 Å), before transforming to the second cubic phase, C_D_. Based on the analysis shown in Figure 2 (left), we conclude that the C_G_ phase, without exception, closely resembled the predicted swelling of a gyroid IPMS with an Ia3d space group symmetry and genus 5. It is further noteworthy that the C_D_ phase also swelled significantly upon DOTAP addition, from covering only a narrow band (from a maximum swelling of about 38% (*w*/*w*) water corresponding to *a* = 93.5 Å) and coexisting with excess water in the pure system, to spanning from 45 to about 70% (*w*/*w*) water (114.3 Å < *a* < 214.4 Å) with a GMO/DOTAP ratio of 80/20 (*w*/*w*) before transforming into the third cubic phase, the primitive C_P_ (Im3m). A small amount of DOTAP (i.e., GMO/DOTAP 97.5/2.5) had already generated a phase transition from C_D_ to C_P_ and caused the C_D_ phase to swell from a maximum 38% (*w*/*w*) water in the pure system up to 45% (*w*/*w*). Referring to the analysis provided in Figure 2 (left), it is evident that experimental data on swelling of the C_D_ phase overlapped well with the predicted curve for a diamond IPMS with a Pn3m space group symmetry and genus 2. On a few occasions, the experimentally determined lattice parameter deviated from the predicted curve, which was indicative of the limited swelling of the C_D_ phase and could be used to estimate the phase boundaries. The C_P_ phase first appeared at about 45% (*w*/*w*) water and then only as a very narrow band with a limited swelling in equilibrium with excess water at a GMO/DOTAP ratio of 97.5/2.5. When increasing the amount of DOTAP (to GMO/DOTAP 95/5), the C_P_ phase swelled further to about 50 wt% water; at GMO/DOTAP 90/10, it comprised from approximately 45 to about 80% (*w*/*w*) water; and at GMO/DOTAP 80/20, it appeared to take close to 90% (*w*/*w*) water before transitioning into a lamellar (L_α_) phase (Appendix A). The presence of vesicles at 95% (*w*/*w*) water content was confirmed by PLOM.

Some further interesting features can be extracted from the analyses provided in Figure 2 (left). First, we observed the co-existence of C_D_ and C_P_ at GMO/DOTAP 97.5/2.5 and different sample water contents (45% (*w*/*w*) and 60% (*w*/*w*)), while there was no sign of the C_D_ at higher water contents, indicating a narrow three-phase region (C_D_+C_P_+water) close to the base line of our phase diagram. Evidently, small variations in GMO/DOTAP-ratios, originating from sample preparations, could determine if the sample were in this three-phase region or in the neighboring two-phase region comprising slightly more DOTAP. Second, the small dip in the C_P_ lattice parameter seen at high water content (>90% (*w*/*w*)) for these samples, which was more pronounced for GMO/DOTAP 95/5 at the same hydration levels, could indicate that the tie lines in this two-phase region (C_P_+water) were not pointing toward the water corner but were rather more horizontal and parallel to the base line. Another explanation of this observation becomes plausible if we analyze the swelling of C_P_ comprising the higher amounts of DOTAP (Figure 2 (left)). When experimental data on the swelling of the C_P_ phase are expected to coincide with the predicted curve for a primitive IPMS with an Im3m space group symmetry and genus 3, the swelling laws we used here assume that the lipid monolayer thickness, l, is constant and equal to that adopted for GMO alone, 17 Å [42]. If appending DOTAP to the system, especially at high water contents, an increase in the average lipid headgroup area would occur due to the excessive hydration of the charged DOTAP headgroup; then, the lipid monolayer thickness would shrink, and the *a*/l used for our calculations would become too small. Indeed, for GMO/DOTAP 80/20, in particular, we see that experimental data were located below the calculated swelling curve. If this is the explanation to our observations, then it would also induce some uncertainty to the swelling limits determined at higher water contents.

### 3.2. Matrix Interactions with Skin

It is a well-accepted fact that skin permeability can be facilitated by occlusion, leading to increased hydration, and the relation between the degree of skin hydration and its permeability to various polar and non-polar compounds has been studied in quite some detail, both at our lab and elsewhere [20,53]. Therefore, it is also recommended that a matrix intended for optimal non-invasive topical extraction of biomarkers comprises a relatively high amount of water and possesses a high inherent water activity to prevent dehydration of the tissue. Based on the obtained phase diagram (Figure 1), we decided to proceed with investigating two candidate matrices, a fully hydrated GMO-water C_D_-phase with 38% (*w*/*w*) water and a GMO/DOTAP/H_2_O C_P_-phase, comprising a GMO/DOTAP ratio of 90/10 and 60% (*w*/*w*) water.

Water sorption isotherms of pure GMO and DOTAP, as well as of GMO/DOTAP 90/10 (*w*/*w*), were obtained at 25 °C with humidity scanning (HS) QCM-D to further characterize their lyotropic phase behavior, as shown in Figure 3. The water uptake by GMO alone revealed the expected phase transitions from L_2_ to L_α_ around a_w_ 0.60–0.70 (i.e., 3–4% (*w*/*w*) water), and then from L_α_ to bicontinuous cubic phases (first C_G_) at a_w_ 0.96–0.98 (i.e., 15–16% (*w*/*w*) water), which is in good agreement with similar data previously reported by Björklund and Kocherbitov [41]. The water sorption of GMO/DOTAP 90/10 (*w*/*w*) was evidently very similar to that of pure GMO with respect to a_w_ values where phase transitions occurred. Further, the phase transitions from C_G_ to C_D_ and C_P_, which should occur very close to a_w_ equal to 1, were not observed by the measurements performed in this work as they were terminated before reaching a_w_ ≈ 1. Instead, complimentary determinations of the a_w_ at 25 °C of the specific lipid–water compositions were made using a water activity meter. Pure GMO with 15% (*w*/*w*) water (L_α_) showed an a_w_ of 0.900, while GMO/DOTAP 90/10 (*w*/*w*) with 30 (C_G_), 45 (C_D_+C_P_), and 60 (C_P_) %(*w*/*w*) water had an a_w_ corresponding to 0.962, 0.991, and 0.999, respectively.

To take this one step further and investigate the effect of topical application of such matrix, we performed a two-hour experiment during a parallel in vivo study on 35 healthy test subjects, which was approved by the Swedish Ethical Review Agency (Dnr 2020-04943). For a more detailed description of the in vivo study, refer to reference [54]. In this in vivo study, we compared the capacity of four alternative matrices with different colloidal properties for the non-invasive sampling of low-molecular-weight biomarkers. To validate the stability of the chosen lipid-based matrices, SAXD measurements were performed after the two-hour application, the results of which are shown in Figure 4. The fully swollen GMO (Pn3m phase), which contained approximately 38 wt% water, and GMO/DOTAP 90/10 (*w*/*w*), containing 60% (*w*/*w*) water (Im3m phase), showed only a minor decrease of the lattice parameter in the case of the GMO/DOTAP matrix (the lattice parameter, *a*, decreased by 8 Å, from 186.6 to 178.7 Å). However, a more pronounced change was observed with the GMO matrix, where the application on skin caused a phase transition from Pn3m (*a* = 94.3 Å) to Ia3d (*a* = 133.6 Å). These effects were most likely caused by the absorption of water from the matrices by the skin. Again, relying on the swelling laws used in Figure 2 above, we calculated the corresponding amounts of water to 2.1 and 7.0 mg·cm^−2^, respectively. This correlates well with the expected water uptake by the skin to reach full hydration at from normal ambient conditions of about 30–40% RH [55,56] and has only a marginal effect on the intended application of these matrices, which is further discussed below.

### 3.3. Sampling for Analysis

Furthermore, the extensive swelling induced by appending a charged lipid to the GMO-water system offers an interesting opportunity, as aiming for a matrix to be used for non-invasive topical extraction of endogenous biomarker molecules also means that what is absorbed by the matrix must be quantified in some way, preferably by analyzing the absorbed solutes in a liquid phase. Thus, we decided to investigate the effect on the phase behavior of the GMO/DOTAP system from adding an electrolyte. In this experiment, 150 mM NaCl aqueous solution was added instead of pure water to the dry lipid mixture. The result is summarized in the partial phase diagram provided in Figure 5 (also see Appendix A) based on the collected data in Figure 2 (right). Evidently, the addition of 150 mM NaCl resulted in a significantly decreased swelling of the system due to a more or less complete screening of interfacial charges. No Im3m phase was formed, and an excess of water in equilibrium with a Pn3m phase was present in all samples, exceeding a water content of 45% (*w*/*w*). Thus, dispersing the Im3m cubic phase in an aqueous 150 mM NaCl solution would result in a phase transition to a less swollen Pn3m phase, expelling water that comprises a significant fraction of the absorbed biomarker molecules. This may provide a simple way to extract the absorbed biomarkers from the lipid matrix for further analysis by a suitable analytical method (e.g., HPLC-UV or LC-MS).

### 3.4. Matrix Optimization

Non-invasive topical extraction of endogenous substances from the skin is inevitably linked to the partitioning of the targeted solutes between the skin and the matrix. Octanol-water partitioning coefficients are furthermore known and tabulated for a vast number of substances, where octanol could perhaps be accepted as a rough substitute for skin, and the water part could be represented by a hydrogel matrix applied on the skin. A hydrogel should thus also work well for the extraction of hydrophilic compounds through skin. Both Kyn and Trp have several pKa values, but none close to the physiological pH. The standard octanol/water partitioning further identified both Kyn and Trp as rather hydrophilic substances (logD_o/w_ (pH7.4): –1.9 and –1.1 (D_o/w_ (pH7.4): 0.0 and 0.1, respectively). From looking at their chemical structures, shown in Figure 6, it is evident that they also could be somewhat surface active and prefer to localize at a lipid–water interface. Indeed, the partitioning of Kyn and Trp between a fully swollen Pn3m cubic phase (GMO/water) and water is in favor of the cubic phase (i.e., K_Q/w_: 1.3 and 2.2, respectively), and the more universal entity derived by Engström and coworkers [45], referred to as the bilayer/water partitioning coefficient, was calculated to K_bl/w_: 1.5 and 3.0 for Kyn and Trp, respectively (Figure 6). The large interfacial area between the interconnected polar and apolar domains of the bicontinuous cubic phases is obviously an asset that can be further explored to optimize the extraction of solutes with specific physicochemical properties. Figure 7 (left) illustrates how this interfacial area varied with water content in the GMO/DOTAP system from less than 50,000 Å^2^ to more than 500,000 Å^2^ per unit cell, easily corresponding up to 500 m^2^·cm^−3^. This figure further shows how the K_Q/w_ depended on K_bl/w_, as well as its superiority over a hydrogel-based extraction matrix for very hydrophilic solutes (Figure 7 right).

These findings were also confirmed in the parallel in vivo study, which included 35 healthy test persons, wherein the capacity of four alternative matrices with different colloidal properties was studied. Four matrices, a chitosan-based hydrogel, an agarose-based hydrogel, and the two lipid-based matrices from the current study, were compared for the non-invasive sampling of low-molecular-weight biomarkers (i.e., tryptophan, kynurenine, tyrosine, and phenylalanine) [54]. The lipid-based matrices were twice as effective as the corresponding hydrogels, and the benefit from electrostatic attraction between matrix and solute was also evident as it appeared to compensate for the smaller interfacial area per unit volume at higher water contents.

## 4. Conclusions

We concluded that there was extensive swelling of the GMO-water system and the formation of a third cubic phase (Im3m) in the presence of DOTAP. We also observed that the cubic phases in this system all formed at rather extreme water activities (>0.9). Physiological salt concentrations counteracted the electrostatic effect of swelling and prevented the formation of the Im3m phase, while wearing the matrix on skin in vivo for several hours only induced a marginal decrease in lattice parameters—most probably an effect of water uptake by the skin tissue. Still, the presence of DOTAP at high salt concentrations resulted in the increased swelling of both the Ia3d and the subsequent Pn3m cubic phases.

The standard octanol/water partitioning identified both Kyn and Trp as rather hydrophilic substances (logD_o/w_ (pH 7.4): –1.9 and –1.1 (D_o/w_ (pH 7.4): 0.0 and 0.1 for Kyn and Trp, respectively), while the partitioning between a fully swollen Pn3m cubic phase (GMO/water) and water was more in favor of the cubic phase (K_Q/w_: 1.3 and 2.2 for Kyn and Trp, respectively). This could be attributed to the large interfacial area between the interconnected polar and apolar domains of almost 500 m^2^·cm^−3^ where the bilayer/water partitioning was calculated for K_bl/w_ (1.5 and 3.0 for Kyn and Trp, respectively). In a parallel in vivo study [54], we showed that the addition of DOTAP had a positive effect on the extraction capabilities for Trp and Kyn (as well as other related amino acids), despite the fact that the significant swelling of the Im3m cubic phase and the related increase in interfacial area per unit cell led to a reduction of the interfacial area per unit volume to about 250 m^2^·cm^−3^. This is a strong argument for the fact that electrostatic attraction also plays a vital role in the extraction process. Evidently, bicontinuous cubic liquid crystals constitute a promising and versatile platform for both transdermal drug delivery and non-invasive extraction of endogenous low-molecular-weight biomarkers through the skin, where the interfacial area per unit volume in a matrix, as well as the incorporation of cationic or anionic molecules at the interface, can be used to optimize the delivery or extraction of particular solutes by the matrix.

## 5. Patents

Patent “Lipid patch”, WO2023066990A2 (published 27 April 2023), Engblom et al. [57].

## Figures and Tables

**Figure 1 pharmaceutics-15-02031-f001:**
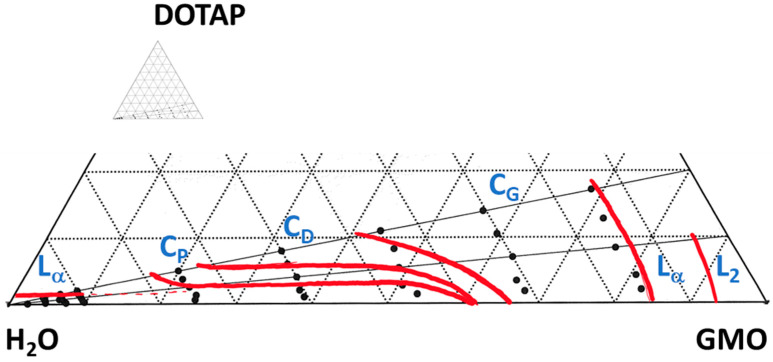
The partial phase diagram of GMO/DOTAP/H_2_O. Black circles represent different phases determined with SAXD. Red solid lines are approximate phase boundaries. Lα—lamellar phase, C_G_—cubic gyroid (Ia3d) phase, C_D_—cubic double diamond (Pn3m) phase, C_P_—primitive cubic (Im3m) phase.

**Figure 2 pharmaceutics-15-02031-f002:**
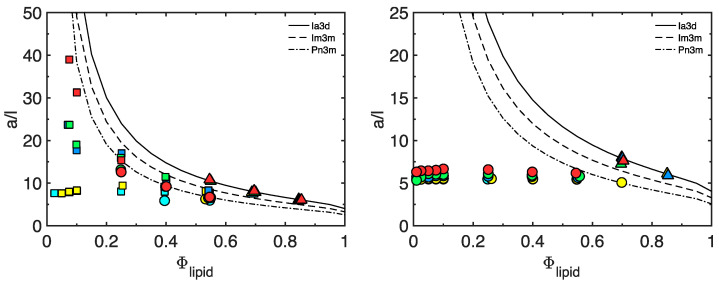
Swelling of the lattice parameter normalized by the monolayer thickness as a function of lipid weight fraction in pure water (**left**) and in 150 mM NaCl solution (**right**). Triangles represent the Gyroid cubic phase (C_G_, space group Ia3d), circles represent the Double Diamond cubic phase (C_D_, space group Pn3m), and squares represent the primitive cubic phase (C_P_, space group Im3m). Color code for GMO/DOTAP composition: cyan—97.5/2.5 (*w*/*w*), yellow—95/5 (*w*/*w*), blue—90/10 (*w*/*w*), green—85/15 (*w*/*w*), and red—80/20 (*w*/*w*).

**Figure 3 pharmaceutics-15-02031-f003:**
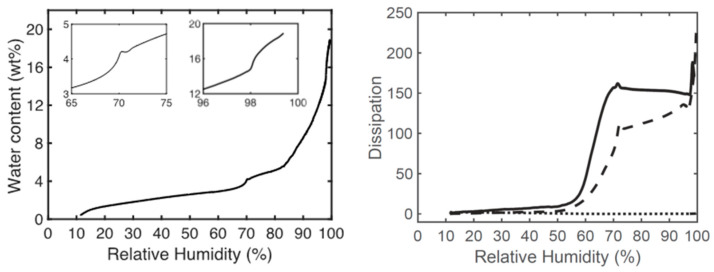
Sorption isotherm of pure GMO (**left**) and dissipation (**right**) as a function of relative humidity for GMO (solid line), DOTAP (dotted line), and GMO/DOTAP 90/10 (*w*/*w*) (dashed line).

**Figure 4 pharmaceutics-15-02031-f004:**
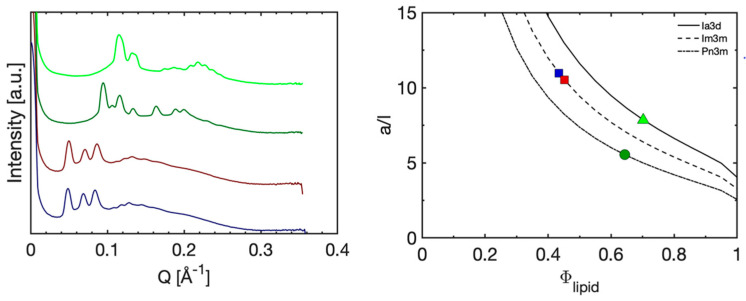
The effect of two-hour in vivo skin application. Diffraction patterns of GMO/DOTAP (90/10 (*w*/*w*)) with 60% (*w*/*w*) water cubic phase (Im3m) measured before (dark violet) and after (dark red), and fully swollen GMO measured before (dark green) and after (light green).

**Figure 5 pharmaceutics-15-02031-f005:**
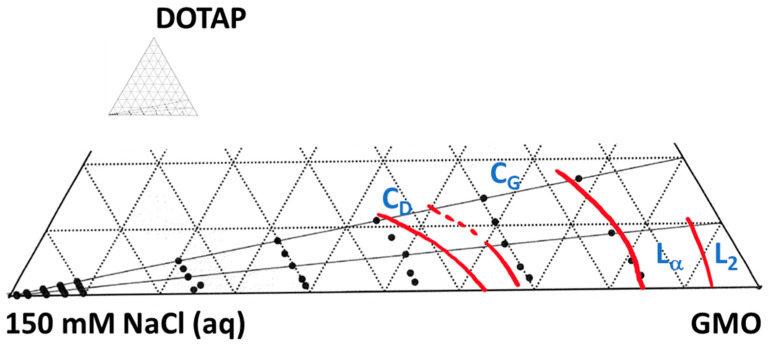
The partial phase diagram of GMO/DOTAP in 150 mM NaCl solution. Black circles represent different phases determined with SAXD. Red solid lines are approximate phase boundaries. L_α_—lamellar phase, C_G_—cubic gyroid (Ia3d) phase, C_D_—cubic double diamond (Pn3m) phase.

**Figure 6 pharmaceutics-15-02031-f006:**
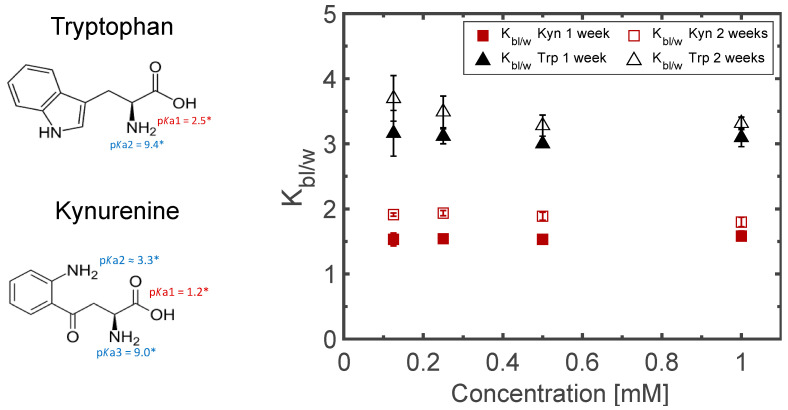
Partitioning of tryptophan (triangles) and kynurenine (squares) into the lipid bilayer of a fully swollen GMO cubic phase (Pn3m). Filled symbols represent partitioning after 1 week, and empty symbols represent partitioning after 2 weeks. * Theoretically determined values using Chemicalize v. 19.7.0, 2019, ChemAxon sofware.

**Figure 7 pharmaceutics-15-02031-f007:**
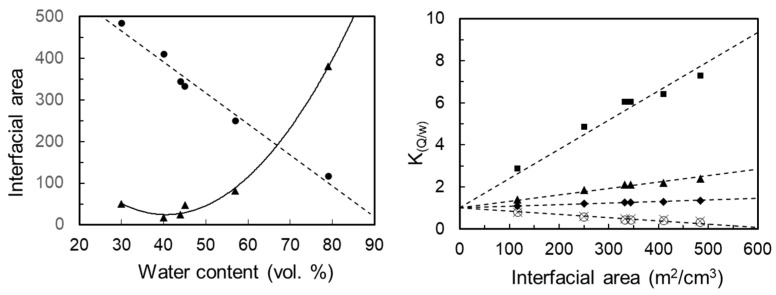
Interfacial area per unit cell (triangles, ×10^3^ Å^2^) and per unit volume (circles, m^2^·cm^−3^). The relationship between K_Q/w_ and interfacial area with respect to K_bl/w_ (10 (squares); 3 (triangles); 1.5 (diamonds); 0.1 (crosses) and 0.01 (circles)).

**Table 1 pharmaceutics-15-02031-t001:** Data show the lipid weight fraction, φ_lipid_; lattice parameter, *a*; normalized lattice parameter by lipid monolayer thickness, *a*/l; radius of the water channel, r; average radii of curvature <R>; area of the unit cell, AUC; length of the water channels, L_w_; volume fractions of lipid, Φ_lipid_; and the symmetry of the phase for pure GMO (30 and 45% (*w*/*w*) H_2_O) and GMO/DOTAP (90/10 (*w*/*w*)) at various water contents.

φ_lipid_	*a*	*a*/l	r	<R>	A_UC_	L_w_	Φ_lipid_	Symmetry
(% *w*/*w*)	(Å)		(Å)	(Å)	(Å^2^)	(Å)	(% *v*/*v*)	
70.2	126.9	7.5	14.5	31.5	49,671	252	73.2	Ia3d
64.0	93.5	5.5	19.5	36.5	16,766	73	64.8	Pn3m
84.3	99.9	5.9	7.8	24.8	30,872	198	87.6	Ia3d
70.0	134.1	7.9	16.2	33.2	55,555	266	70.0	Ia3d
54.2	111.4	6.6	26.5	43.5	23,803	87	55.6	Pn3m
54.2	141.3	8.3	26.2	43.2	46,843	173	56.0	Im3m
40.0	186.6	11.0	40.1	57.0	81,650	228	43.4	Im3m
25.1	289.3	17.0	71.5	88.4	196,196	353	28.5	Im3m
10.0	301.2	17.7	75.2	92.0	212,760	368	27.4	Im3m
10.0	406.4	23.9						L_α_
7.0	402.8	23.7	106.3	123.0	380,471	492	20.6	Im3m
7.0	534.7	31.5						L_α_
5.0	708.8	41.7						L_α_

## Data Availability

All data are presented within the manuscript and Appendix A.

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
