# Peer review of "Bicontinuous Cubic Liquid Crystals as Potential Matrices for Non-Invasive Topical Sampling of Low-Molecular-Weight Biomarkers"

_pharmaceutics, 2023, doi:10.3390/pharmaceutics15082031_

Round 1
Reviewer 1 Report
This manuscript describes the use of bicontinuous cubic liquid crystals as potential matrices for non-invasive topical extraction biomarkers through the skin in many skin disorders, including cancer. The formulation development was rationally carried out and all the experimental details are comprehensively provided. I congratulate the authors for a very interesting work that is very well conducted and the manuscript is clinically relevant and my suggestion is to publish it. There are just a few comments that should be considered to strengthen the manuscript (see below):
1. Glyceryl monooleate (GMO) is an amphiphilic surfactant, which as such can solubilize hydrophilic, lipophilic and amphiphilic molecules in its different polarity regions. Have the authors previously tested the matrices without DOTAP?
2. Is GMO soluble in water? How did you solubilize GMO in water? Have the authors used any other excipient for the matrices preparation? Describe more detailed the preparation method.
3. Generally topical application products require flow and viscosity measurements.
Have the authors considered determining the viscosity of GMO:DOTAP and GMO: DOTAP: NaCl sol matrices? Does authors consider necessary to do it?
4. The authors would have to expand information on the in vivo studies on healthy volunteers. Describe more detailed the in vivo study methodology and include the Approval date of the Ethical Committee (line 520).
5. Are the used HPLC methods validated? It is important to add some information on the validation parameters. Please rephrase and write correctly the analytical method used.
Author Response
Dear Reviewer 1,
Thank you for the kind en constructive comments on our manuscript, please see our point-by-point response in the enclosed file.

Reviewer 2 Report
The research work: Bicontinuous cubic liquid crystals as potential matrices for non-invasive topical sampling of low molecular weight biomarkers” focus on the extraction of Tryptophan and Kynurenine, two compounds associated with several inflammatory skin disorders. They hypothesize that lipid-based bicontinuous cubic liquid crystals could be efficient extraction matrices. The researchers concluded that bicontinuous cubic liquid crystals constitute a promising and versatile platform for non-invasive extraction of biomarkers through skin, as well as for transdermal drug delivery. The work is novel and applicable.
The minor observations and comments are as follows:
1. 2.2. Sample preparation: Please cite if the method was adopted from previous work.
2. 2.3. Small angle X-Ray Diffraction (SAXD): Please cite if method adopted from previous work.
3. L 136: |?| = ? = 4? ? sin ( ? 2 ), this is equation 1… change the following equation numbers…
4. 2.8. HPLC-UV analysis, Cite the method is not developed.
5. 3.1. The matrix, the author can discuss and compare the work previously done specifically, with GMO or any other lipid.
6. Does the prepared matrix stable?
Author Response
Dear Reviewer 2,
Thank you for the kind en constructive comments on our manuscript, please see our point-by-point response in the enclosed file.
